# Alveolar macrophages are critical for broadly-reactive antibody-mediated protection against influenza A virus in mice

Wenqian He[1,2], Chi-Jene Chen[1], Caitlin E. Mullarkey[1], Jennifer R. Hamilton[1,2], Christine K. Wong[1], Paul E. Leon[1,2], Melissa B. Uccellini[1], Veronika Chromikova[1], Carole Henry[3], Kevin W. Hoffman[1], Jean K. Lim[1], Patrick C. Wilson[3], Matthew S. Miller[4], Florian Krammer [1], Peter Palese[1] & Gene S. Tan[1,5]

The aim of candidate universal influenza vaccines is to provide broad protection against influenza A and B viruses. Studies have demonstrated that broadly reactive antibodies require Fc–Fc gamma receptor interactions for optimal protection; however, the innate effector cells responsible for mediating this protection remain largely unknown. Here, we examine the roles of alveolar macrophages, natural killer cells, and neutrophils in antibody-mediated protection. We demonstrate that alveolar macrophages play a dominant role in conferring protection provided by both broadly neutralizing and non-neutralizing antibodies in mice. Our data also reveal the potential mechanisms by which alveolar macrophages mediate protection in vivo, namely antibody-induced inflammation and antibody-dependent cellular phagocytosis. This study highlights the importance of innate effector cells in establishing a broad-spectrum antiviral state, as well as providing a better understanding of how multiple arms of the immune system cooperate to achieve an optimal antiviral response following influenza virus infection or immunization.

[1] Department of Microbiology, Icahn School of Medicine at Mount Sinai, New York, NY 10029, USA. [2] Graduate School of Biomedical Sciences, Icahn School of Medicine at Mount Sinai, New York, NY 10029, USA. [3] Department of Medicine, Section of Rheumatology, The Knapp Center for Lupus and Immunology Research, The Committee on Immunology, The University of Chicago, Chicago, IL 60637, USA. [4] Department of Biochemistry and Biomedical Sciences, Institute of Infectious Diseases Research, McMaster Immunology Research Centre, McMaster University, Hamilton, ON, Canada L8S 4K1. [5] Present address: J. Craig Venter Institute, 4120 Capricorn Lane, La Jolla, CA 92037, USA. Correspondence and requests for materials should be addressed to G.S.T. (email: gtan@jcvi.org)

nfluenza virus, a highly contagious pathogen that causes an acute infection of the upper respiratory tract, is a significant source of public health and socio-economic burden around the world. Despite the availability of vaccines and antiviral therapeutics, influenza virus still causes an estimated 3–5 million cases of severe disease and up to 500,000 deaths annually worldwide[1]. The antigenic changes in the hemagglutinin (HA), the chief target of neutralizing antibodies, and to a lesser extent, in the neuraminidase (NA), provide influenza virus a mechanism by which to evade pre-existing immunity in the human population[2–4]. A yearly reformulation of the influenza virus vaccine is thus required to address the frequent antigenic drift observed on the surface antigens of circulating strains. The modular segmented nature of the influenza virus genome poses an additional problem, allowing for the creation of a novel influenza virus from two or more different strains through antigen shift, which has caused pandemics in the past[2].

Notwithstanding the challenges posed by a highly variable pathogen, the humoral antibody response elicited by current influenza vaccines still provides the best protection against disease. Antibodies that target the receptor-binding site (RBS) located on the HA globular head are the predominant type of antibodies elicited during vaccination and are the main correlates of protection. While there are broadly neutralizing antibodies (bNAbs) that target the head[5], the neutralization breadth of head-specific antibodies is generally limited to highly similar virus strains. In contrast, a minority of antibodies elicited after immunization are ones that recognize the stalk region of the HA. Antibodies that target the stalk are typically broadly neutralizing against a host of divergent strains and different influenza virus subtypes—reflecting the more antigenically conserved portion of the HA molecule.

Inhibition by both types of antibodies, head- or stalk-directed, can be readily measured by conventional in vitro neutralization assays. The hemagglutination-inhibition (HI) assay specifically detects antibodies that target the RBS, while neutralization assays can be utilized to determine the antiviral activities of both head- and stalk-specific antibodies. Protection from influenza virus, however, consists of more than neutralizing antibodies. As suggested by several studies in humans, the presence of non-neutralizing antibodies (nonNAbs), which are undetectable by traditional in vitro neutralization assays, may contribute to the protection against overt disease[6–8]. Nevertheless, the target(s) of nonNAbs elicited by influenza vaccination or infection and the mechanisms by which they protect have yet to be fully elucidated. Several studies recently reported protection against disease in mice with HA-specific monoclonal antibodies (mAbs) lacking in vitro neutralizing activity[9–11]. Otherwise, most of the data concerning the role of nonNAbs in protection have been generated in candidate vaccines that target two viral proteins: the ectodomain of the proton pump channel, M2, and the nucleoprotein (NP)[12, 13].

The aim of this study is to characterize the mechanisms by which broadly reactive antibodies that target HA confer protection in vivo. Previous studies have demonstrated that bNAbs that target the stalk region of the HA require Fc–Fc gamma receptor (FcγR) interactions for optimal protection in vivo[9, 14]. In light of these findings, we seek to identify the main innate immune cells that are responsible for clearing influenza A virus (IAV) infection and the mechanisms by which they contribute to antibody-mediated protection. Here, we demonstrate that both murine and human nonNAbs that target the HA require alveolar macrophages (AMφ), and are dependent on Fc–FcγR interactions to optimally protect mice from gross morbidity and death. Furthermore, we also show that bNAbs require AMφ to mediate optimal protection. Potent activation of AMφ by these antibodies to release pro-inflammatory molecules and phagocytose immune complexes may be involved in the optimal clearance of virus infection in vivo. The current study highlights the importance of AMφ in mediating the antibody-dependent responses against influenza virus infection. Vaccination strategies that boost nonNAb responses in addition to eliciting robust neutralizing antibody titers may be particularly effective in providing protection against influenza virus infection. Moreover, such protective nonNAb titers should be taken into account when evaluating the efficacy of a given vaccine.

## Results

**NonNAbs protect mice from lethal IAV challenge.** A 6–8-week-old female BALB/c mouse was sequentially infected with sublethal doses of three divergent H3N2 viruses (details in "Methods" section). Three weeks after the last infection, the mouse was boosted intraperitoneally (IP) with 100 μg of a purified preparation of A/Perth/16/09 (H3N2) virus; three days after the boost, the spleen was collected and fusion with SP2/0 myeloma cell line was performed. Monoclonal hybridoma cultures secreting antibodies were initially screened for binding against HK/68 H3 HA by ELISA. We identified three mAbs that are broadly reactive to H3 HAs as indicated by ELISA (Supplementary Table 1): mAbs 2B9, 2C10, and FEE8. No detectable binding was observed against H4 (A/duck/Czechoslovakia/1956), H7 (A/Shanghai/02/2013), and H10 (A/mallard/Interior Alaska/10BM01929/2010). The murine mAb 9H10 is a previously characterized bNAb that targets the stalk region of H3 and H10 HAs[15], serving as a positive control here. The binding affinity of each mAb to A/Hong Kong/1/68 (HK/68), A/Victoria/316/11 (Vic/11), and A/Philippines/1/82 (Phil/82) H3 HAs were then determined using biolayer interferometry (Table 1). Compared to bNAb 9H10, the affinities of 2B9, 2C10, and FEE8 were 100–1000 fold lower. Their neutralization activity was examined by microneutralization assays and HI assays, and none exhibited neutralization activity against X31 (a reassortant virus with the HA and NA of HK/68 and the internal proteins of A/Puerto Rico/8/34 (PR/8)) (Fig. 1a and Supplementary Fig. 1A). They all recognized the HA1 domain (most of which comprises the HA head) of the H3 HA glycoprotein as shown by western blot using trypsin-treated HK/68 H3 proteins resolved under non-reducing gel conditions (Fig. 1b). Interestingly, these broadly reactive nonNAbs were able to fully protect mice from lethal challenge of X31 at a dose of 15 mg per kg (Fig. 1c, d).

### Table 1 Affinity measurements of monoclonal antibodies by biolayer interferometry

| Subtype | Isolate | 9H10 | 2B9 | 2C10 | FEE8 |
| --- | --- | --- | --- | --- | --- |
| | | $K_d$ (M) | | | |
| H3N2 | A/HongKong/1/1968 | $(2.14 \pm 0.6)E{-}11$ | $(2.92 \pm 0.04)E{-}8$ | $(1.22 \pm 0.02)E{-}8$ | $(1.64 \pm 0.3)E{-}9$ |
| H3N2 | A/Victoria/361/2011 | $(4.99 \pm 0.4)E{-}11$ | $(1.95 \pm 0.8)E{-}9$ | $(3.65 \pm 0.1)E{-}8$ | $(3.84 \pm 0.3)E{-}8$ |
| H3N2 | A/Philippines/2/1982 | $(6.72 \pm 0.33)E{-}11$ | $(2.88 \pm 0.9)E{-}9$ | $(2.31 \pm 0.05)E{-}8$ | $(1.55 \pm 0.05)E{-}8$ |

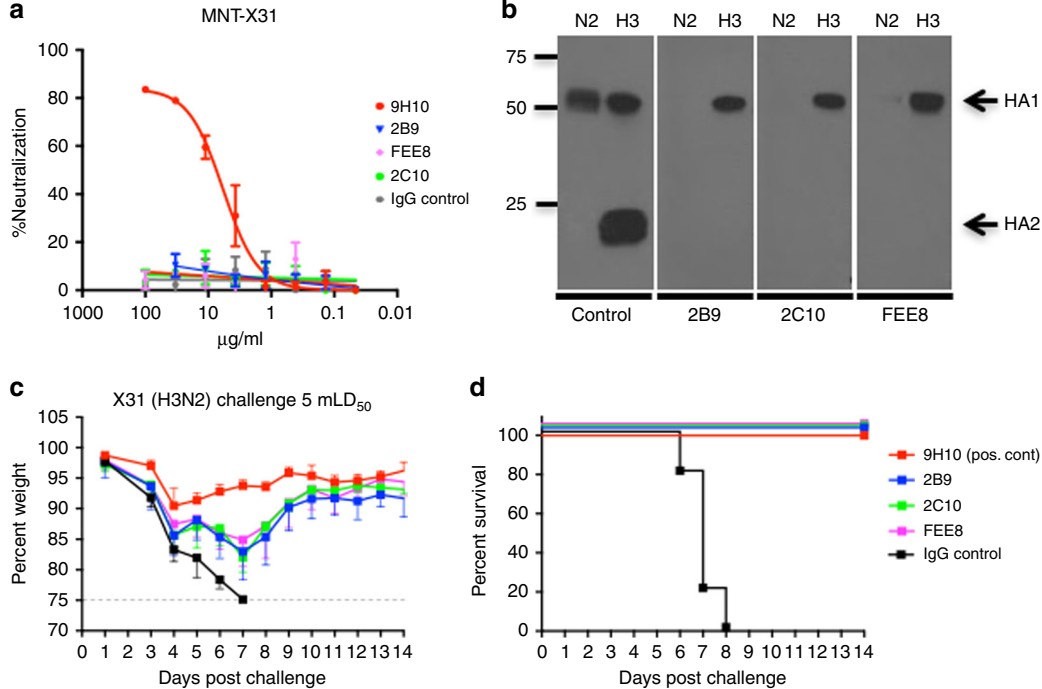

**Fig. 1** Murine HA-specific nonNAbs can protect mice from lethal IAV challenge. **a** The neutralization activities of 2B9, 2C10, and FEE8 were tested against X31 virus by microneutralization assay on MDCK cells. Values represent mean ± SD of three independent experiments. **b** A western blot was performed against recombinant HK/68 H3 (cleaved by trypsin) and N2 protein using mAb 2B9, 2C10, or FEE8. For controls, mAbs XY102 and 12D1 were used to detect the HA1 and HA2 of HK/68 HA, respectively. An anti-histidine mAb was used to detect HK/68 NA. **c, d** The ability of nonNAbs to confer protection was tested against X31 infection in vivo. Groups of BALB/c mice ($n = 5$) were injected with 15 mg per kg IP of each indicated mAb and then challenged intranasally (IN) with 5 mouse 50% lethal doses ($mLD_{50}$) of X31. Control IgG (22A6) is a murine mAb that recognizes glutathione S-transferase (GST). **c** The change in weights and (**d**) the survival curves are shown through a 14-day period. Values represent mean ± SD

**NonNAbs induce high-inflammatory responses in lungs.** We next assessed the inflammatory status in the lungs from mice treated with 2B9 or FEE8 during IAV infection. At 3 and 6 days post infection (dpi), lungs were collected, sectioned, and stained with hematoxylin and eosin (Fig. 2a). Although all groups exhibited inflammation in the lungs compared to naive mice, we observed an increase in alveolar infiltration in 2B9 and FEE8-treated animals compared to 9H10 or IgG group at 3 and 6 dpi (Fig. 2b). Of note, while 9H10- and IgG-treated mice have comparable histological pathologies on days 3 and 6, we speculate that the similarities are due to different reasons. In the first scenario, we hypothesize that a high concentration of bNAb 9H10 allows for the clearance of virus without the engagement of Fc-mediated responses and fails to trigger an inflammatory response. In the latter, influenza virus infection alone may contribute to the transient suppression of the innate immune response.

To investigate the effect of nonNAb-mediated inflammation during an acute influenza virus infection, we collected the bronchoalveolar lavage fluid (BALF) to measure the cytokine and chemokine profile. We found that at 3 dpi, 14 out of 23 proteins tested were detected at higher levels in 2B9- or FEE8-treated mice than control IgG-treated ones, suggesting that protective nonNAbs induce high alveolar inflammatory responses in vivo during IAV infection (Fig. 2c). Protein levels for the 14 cytokines/chemokines are shown in Supplementary Fig. 2A. We did not detect any substantial differences in the inflammatory profile between the groups at 6 dpi, which may due to the short half-life of these inflammatory molecules[16–18]. Taken together, our results demonstrate that protective nonNAbs can induce a protective inflammatory response, which may help clear viruses.

**NonNAbs require AMφ for mediating protection.** During infection or vaccination, the protective polyclonal antibody response is composed of both neutralizing and nonNAbs[10, 19]. While the antiviral activity of neutralizing antibodies is easily measured by established in vitro assays, nonNAbs are more difficult to specifically detect, making their potential contributions to in vivo protection easy to overlook. Since these antibodies are elicited in individuals who have been previously exposed to IAV, it is important to study the mechanisms by which they confer protection in vivo. Broadly reactive non-neutralizing HA antibodies have been shown to require Fc–FcγR interactions for optimal protection in vivo[10, 11]. However, the contributions of the different innate immune cell populations responsible for cooperating with nonNAbs in protection remain elusive. Studies have shown that AMφ play a critical role in limiting IAV replication and spread after infection[20–22]. AMφ is a type of Mφ that reside in the alveolar region of the lower respiratory tract[23]. To investigate whether AMφ can cooperate with nonNAbs to confer protection, we depleted AMφ in mice at day −2 and day 0 using clodronate liposome delivered intranasally. A group of mice was given PBS-liposome as a control group. On day 0, mice were challenged with a lethal dose of X31 virus 2 h after intraperitoneal administration of mAbs. Consistent with previous results, mice receiving 2C10, 2B9, or FEE8 without depletion treatment fully survived the lethal challenge (Fig. 3b–d). However, mice depleted of AMφ all exhibited severe morbidity and lost protection except for one survivor. It should also be noted that nonNAb-treated mice depleted of AMφ failed to induce an inflammatory response, which may contribute to substantial morbidity and mortality (Supplementary Fig. 2B). This observation further indicates that AMφ are central in eliciting mAb-mediated inflammatory

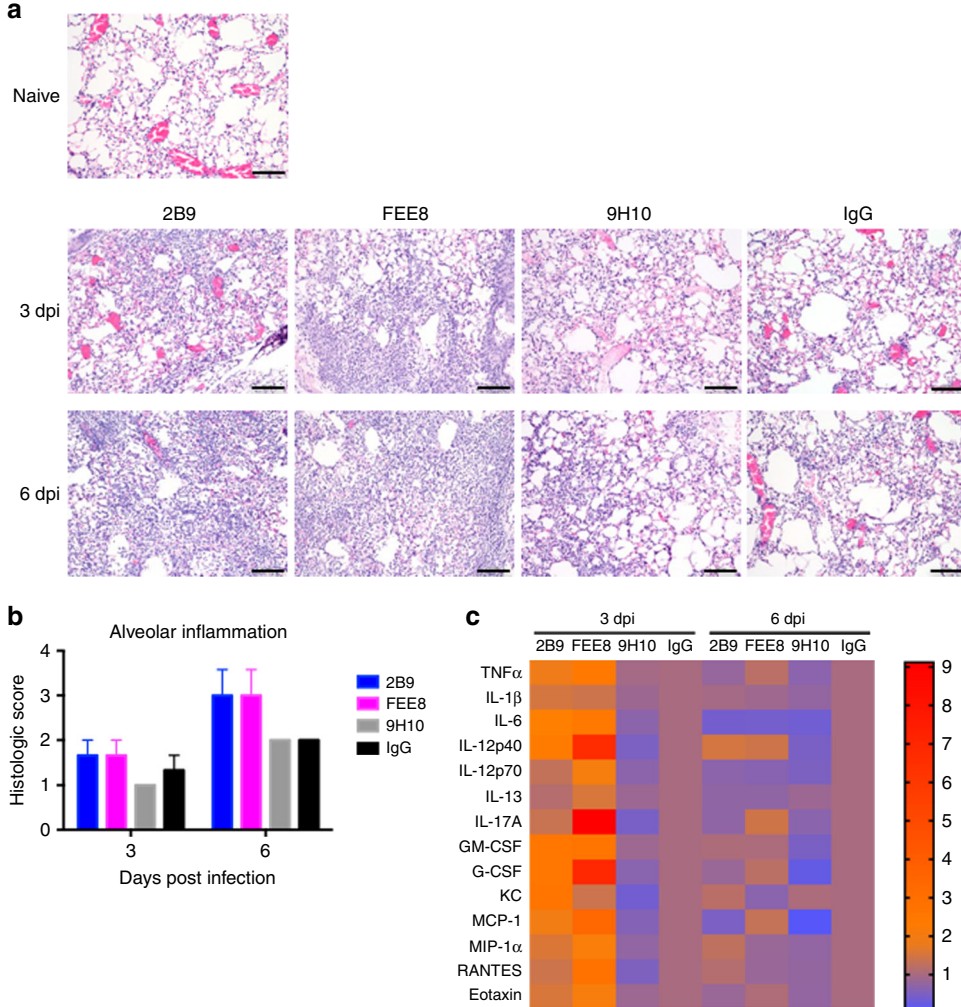

**Fig. 2** Mice administered with nonNAbs have higher alveolar inflammatory responses upon IAV infection. Groups of BALB/c mice ($n = 3$) were administered IP with indicated mAbs (15 mg per kg) and challenged IN with 5 $mLD_{50}$ of X31. **a** At 3 and 6 dpi, lung sections from naive, 2B9, FEE8, 9H10, or IgG-treated animals ($n = 3$ mice per group) were stained with hematoxylin and eosin. *Scale bar*, 100 μm. Control IgG is a murine mAb that targets GST. **b** Independent pathology quantification of alveolar inflammation is as indicated. *Error bars* represent standard error mean (SEM). **c** BALF were also collected from animals and analyzed for cytokine/chemokine production. A heat map was generated using GraphPad Prism. Heat map represents fold changes of cytokine/chemokine protein levels in BALF from the indicated treatments. Values are normalized to corresponding IgG controls

responses required for protection. The group that received the bNAb 9H10 displayed no significant difference in weight loss and survival between depleted and non-depleted animals (Fig. 3a); it is expected that a high concentration of a stalk-specific bNAb protects independently of Fc–FcγR interactions[14]. Since nonNAbs 2B9 and FEE8 are broadly reactive H3 HA antibodies, we tested the protective efficacy of 2B9 and FEE8 against a lethal challenge of X79 virus (H3N2). While there is a difference in the kinetics of morbidity and mortality in control IgG-treated mice against X79, (Supplementary Fig. 3A, B), our data still indicate a requirement for AMφ and nonNAb in heterologous protection.

Our previous data demonstrated that high-affinity nonNAbs (~100-fold higher affinity than FEE8) can provide protection against lethal challenge down to 1 mg per kg with negligible weight loss[11]—suggesting that affinity plays a role in protection. To determine if high-affinity nonNAbs require AMφ for protection, 6- to 8-week old BALB/c were depleted of AMφ as previously described and administrated with 1 mg per kg of nonNAb 1H5[11] IP before a challenge with a lethal dose of H7N9 influenza A virus. In Supplementary Fig. 3C, we demonstrated

that mice treated with nonNAb 1H5 required the presence of AMφ for protection.

At 3 and 6 dpi, about 95% of AMφ in the lungs were depleted in clodronate-treated mice as analyzed by flow cytometry (Supplementary Fig. 4A). IAV infection can result in reduction of AMφ when comparing PBS-liposome-treated and naive animals, which has been shown by previous studies[24]. Monocytes, neutrophils, and two subsets of pulmonary dendritic cells (CD103+CD11b$^{low}$ and CD103−CD11b$^{high}$) in the lungs were not affected by the clodronate treatment (Supplementary Fig. 4B–E), demonstrating that our clodronate treatment selectively depleted AMφ. When 15 mg per kg of mAb 2B9, 2C10, or FEE8 was passively transferred into mice on day 0, we also looked at the number of AMφ in the lung and BALF in mice upon antibody treatment. We found that in non-depleted mice, administration with nonNAb and bNAb prevented the depletion of AMφ at 6 dpi when compared to IgG control-treated mice (Supplementary Fig. 4F, G).

To determine if protection and survival correlated with lower viral replication, we collected lungs from each group at 3 and 6 dpi, and measured their viral lung titers by plaque assays. At

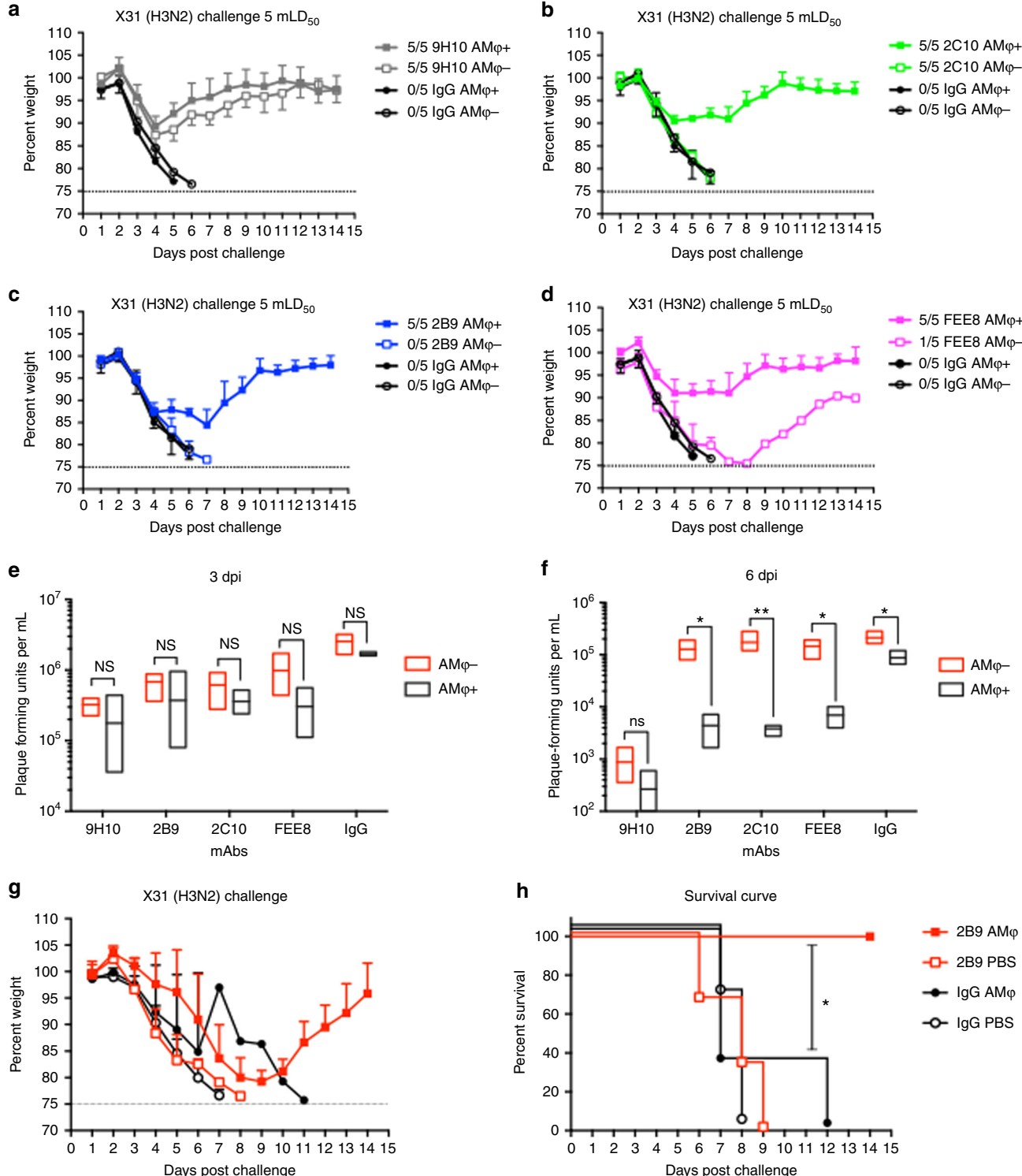

**Fig. 3** Murine nonNAbs require AMφ to confer protection in vivo. BALB/c mice (*n* = 5) were depleted of AMφ using clodronate liposome administered IN. PBS-liposome-treated mice were used as controls. Mice were injected IP with 15 mg per kg: (**a**) 9H10, (**b**) 2C10, (**c**) 2B9, or (**d**) FEE8 5 h before a challenge with 5 mLD$_{50}$ of X31 virus. Weight change was monitored daily. The ratios in the figure legends indicate the number of animals that survived challenge over total number of animals per group. A mAb against GST was used as an IgG control. At (**e**) 3 and (**f**) 6 dpi, viral lung titers were determined by plaque assays. **g**, **h** GM-CSF knockout mice were reconstituted with 200,000 cells per 50 μL of wild-type AMφ collected from C57BL/6J mice. Twenty-four hours post adoptive transfer, these mice (*n* = 3–4 mice per group) were administered with 15 mg per kg of mAb 2B9 or IgG control. Two hours post administration of mAbs, mice were challenged with a lethal dose of X31. The (**g**) weights and (**h**) survival were monitored for 14 days. Statistical significance was determined using two-way ANOVA and Sidak's multiple comparisons tests (GraphPad Prism). For all panels: *$P \leq 0.05$, **$P \leq 0.01$, not significant (NS). Values represent mean ± SD

3 dpi, no significant differences were observed between depleted and non-depleted animals (Fig. 3e). However, at 6 dpi, viral titers were significantly higher in depleted animals than non-depleted

groups, indicating that AMφ in addition to nonNAbs are required to control viral replication in the lungs at 6 dpi (Fig. 3f). No significant difference in the viral lung titers was expected in

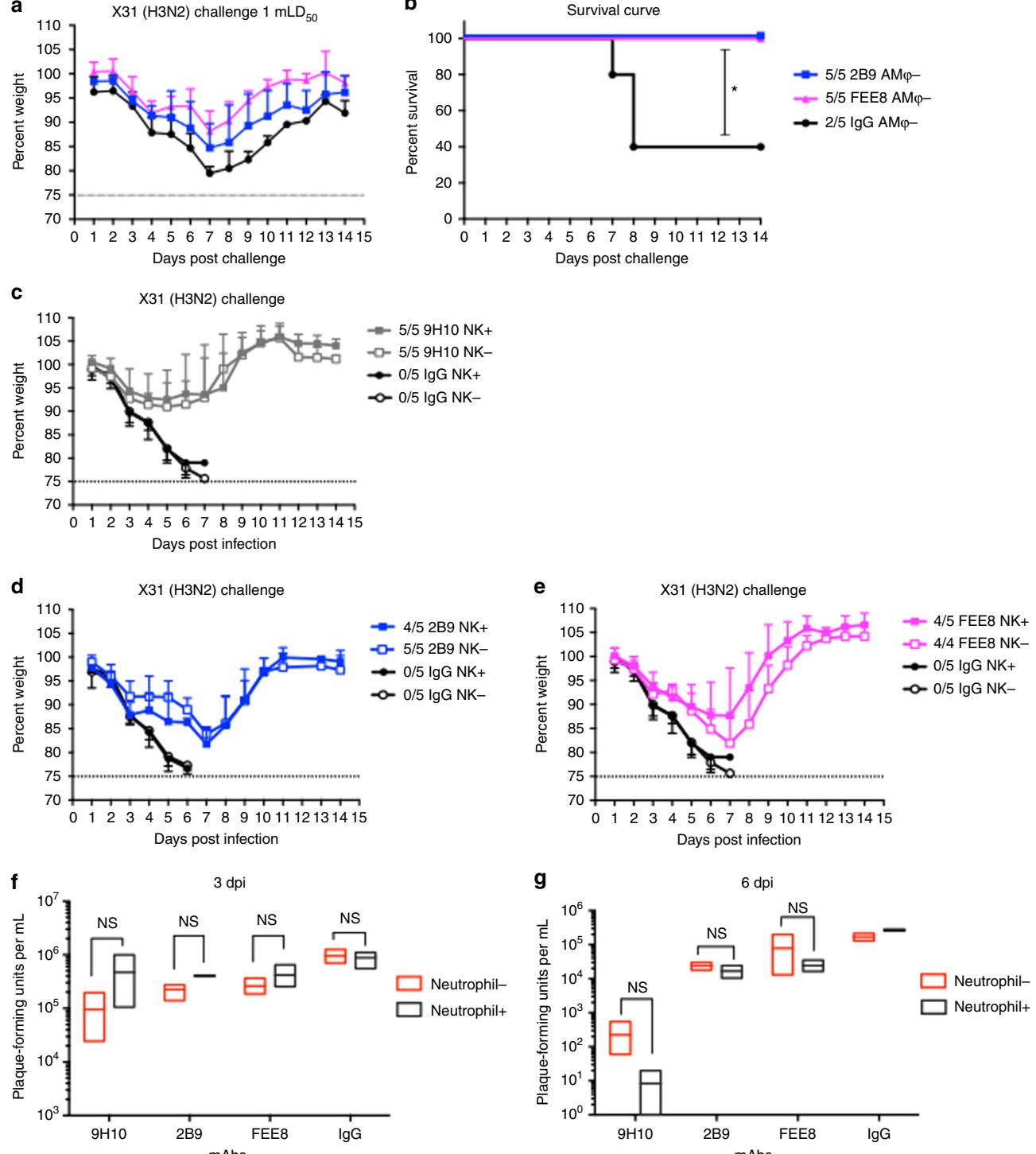

**Fig. 4** NK cells and neutrophils do not play significant roles in murine nonNAb-mediated protection. **a, b** BALB/c mice depleted of AMφ were administered IP with 2B9, FEE8, or IgG, and then challenged IN with 1 mLD$_{50}$ of X31 virus. **c–e** C57BL/6J mice were treated IP with mAb PK136 to deplete NK1.1 cells. Control mice received PBS. Mice received 15 mg per kg of (**c**) 9H10, (**d**) 2B9, or (**e**) FEE8 IP 5 hours before a challenge with 5 mLD$_{50}$ of X31. The ratios in the figure legends indicate the number of animals that survived challenge over total number of animals per group (n = 4 or 5). A mAb against GST was used as an IgG control. **f, g** BALB/c mice were treated with mAb 1A8 to deplete neutrophils. Control mice received PBS. The mice were given 9H10, 2B9, or FEE8 at 15 mg per kg. At (**f**) 3 and (**g**) 6 dpi, viral lung titers were determined by plaque assays. Two-way ANOVA and Sidak's multiple comparisons tests were used to determine statistical significance (GraphPad Prism) (n = 3). Values represent mean ± SD. For all panels: *P ≤ 0.05, **P ≤ 0.01, not significant (NS)

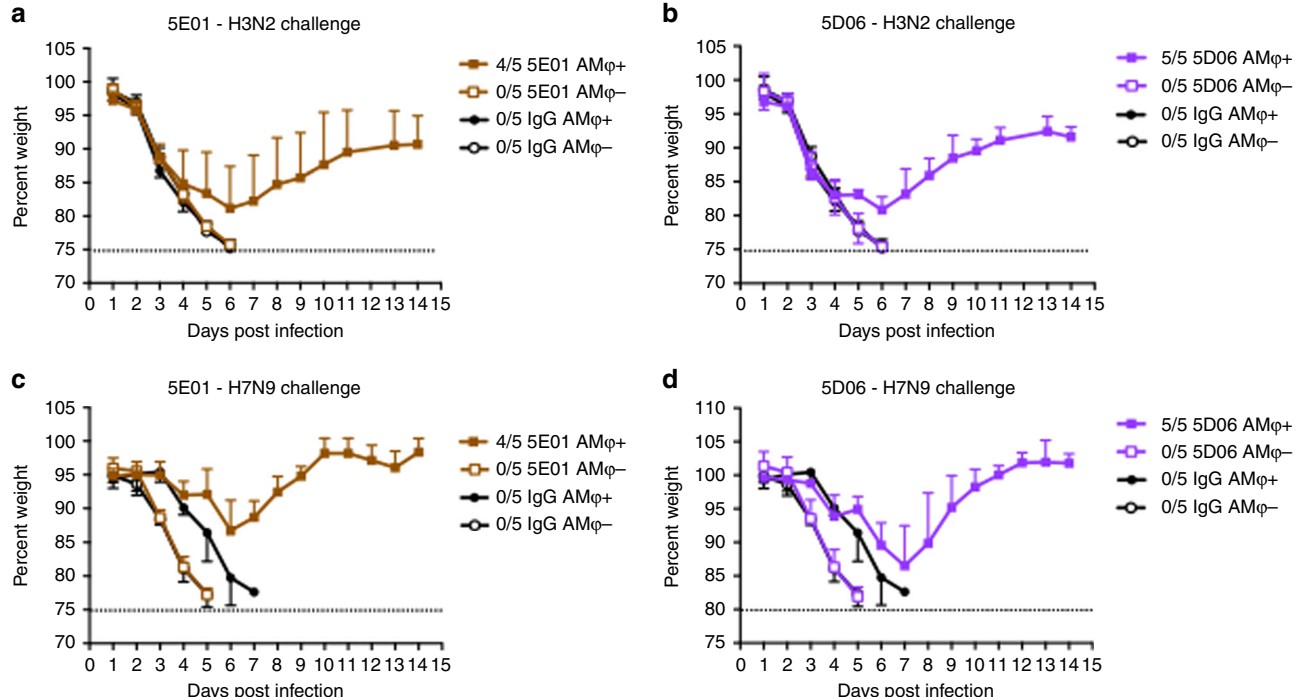

**Fig. 5** Heterosubtypic protection provided by human nonNAbs requires AMφ. BALB/c mice were treated IN with clodronate liposome to deplete AMφ. PBS liposome was used in non-depleted groups. Mice received 6 mg per kg (**a**) 5E01 or (**b**) 5D06 IP, and then were challenged IN with 5 mLD$_{50}$ of X31 (H3N2). Another set of mice were administered IP with 3 mg per kg (**c**) 5E01 or 2 mg per kg (**d**) 5D06, and then challenged IN with 10 mLD$_{50}$ of A/Shanghai/1/13 (H7N9). The ratios in the figure legends indicate the number of animals that survived challenge over total number of animals per group. Values represent mean ± SD

9H10-treated mice, since at high concentrations, stalk-specific bNAbs primarily rely on their neutralizing activity rather than Fc–FcγR interaction for protection.

To further assess the role of AMφ in Fc-mediated immunity, we adoptively transferred wild-type AMφ into granulocyte macrophage-colony stimulating factor (GM-CSF) knockout mice, which have been shown to have severely impaired AMφ—lacking in their ability to undergo phagocytosis and produce a proper inflammatory response[25]. Female GM-CSF knockout mice were initially reconstituted with 200,000 AMφ from wild-type naive female C57BL/6J or with 1× PBS. Twenty-four hours post adoptive transfer, mice were administered with either nonNAb 2B9 or IgG control, and subsequently challenged with X31 virus. As indicated in Fig. 3g, h, 2B9-treated GM-CSF knockout mice that received wild-type AMφ were fully protected from lethal challenge, whereas mice that received PBS died. As expected, IgG-control-treated GM-CSF knockout mice that received either wild-type AMφ or PBS all succumbed to infection. Of note, confirmation of successful adoptive transfer with wild-type AMφ into GM-CSF knockout mice is shown in Supplementary Fig. 5. Here, we demonstrate that the presence of functional AMφ are required for in vivo protection by nonNAbs.

**Natural killer cells and neutrophils play minor roles in protection.** Upon IAV infection, multiple innate immune cell populations are recruited to the lung to combat viral infection, such as natural killer (NK) cells, neutrophils, and monocytes. Besides that, complement pathway can be activated to kill virus-infected cells as well. To determine whether mechanisms independent of AMφ also contribute to nonNAb-mediated protection, we performed a challenge study in AMφ-depleted mice. These mice were given a sublethal dose (1 mLD$_{50}$) of X31 virus.

IgG control group underwent severe weight loss and only 40% of the mice survived, whereas mAb 2B9 and FEE8 were able to fully protect the mice from infection (Fig. 4a, b). After depletion of AMφ, the protection mediated by mAb 2B9 or FEE8 was not completely impaired, suggesting that mechanisms independent of AMφ also contributed to the protection.

We next sought to investigate whether NK cells play a role in the observed protection by nonNAbs. Within days of influenza virus infection, NK cells can be recruited to the lung and activated to kill virus-infected cells by releasing cytotoxic granules via an antibody-dependent manner[26]. We utilized the mAb PK136 to deplete NK cells in C57BL/6J mice, which has been described previously[27]. Of note, mAb PK136 has been previously shown to work robustly in certain strains of mice such as C57BL/6J, SJL, and NZB, but is ineffective against other strains due to allelic variation in the NK cell receptor protein 1 gene that encodes CD161 (NK1.1)[28, 29]. The mice received PK136 on day −2, 0 (day of infection), and 5. Another group received PBS as control. Depletion efficacy was confirmed by flow cytometry (Supplementary Fig. 6A, B). Upon X31 challenge, mice receiving nonNAbs exhibited no significant differences in weight loss and survival between NK-depleted and non-depleted groups (Fig. 4c–e), suggesting that NK cells do not play a significant role in nonNAb-mediated protection.

Neutrophils are the first innate immune cell population recruited to the site of infection. They play a critical role in controlling infection via secreting cytokines and chemokines, engulfing antibody-opsonized pathogens, and forming neutrophil extracellular traps[30]. We utilized the mAb 1A8 to deplete neutrophils in BALB/c mice[31]. Depletion efficacy was confirmed by flow cytometry (Supplementary Fig. 7A, B). Mice were treated IP with mAb 1A8 on day −1, 1, 3, and 5 (day 0 is the day of challenge). Mice that received PBS served as controls. On day 0,

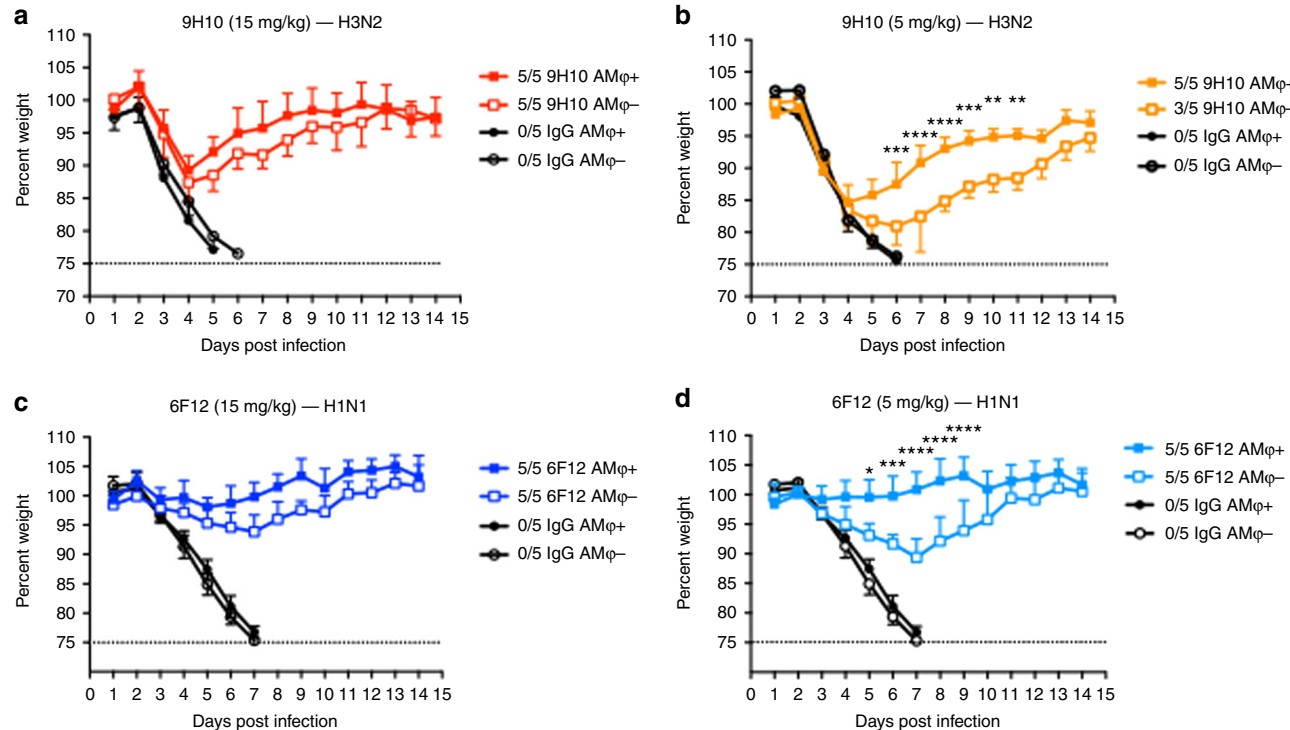

**Fig. 6** AMφ play an important role in mediating in vivo protection by bNAb. BALB/c mice were treated IN with clodronate liposome to deplete AMφ. Control mice received PBS liposome. Mice were either treated with (**a**) 15 mg per kg or (**b**) 5 mg per kg of 9H10 IP, and were challenged IN with 5 mLD$_{50}$ of X31. Another set of mice received (**c**) 15 mg per kg or (**d**) 5 mg per kg of 6F12, and then challenged with 5 mLD$_{50}$ of NL09. Of note, the (**a**) 15 mg per kg data set is identical to the AMφ + group in Fig. 3a. A mAb against GST was used as an IgG control. The ratios in the figure legends indicate the number of animals that survived challenge over total number of animals per group. Two-way ANOVA and Sidak's multiple comparisons tests were used to determine statistical significance (GraphPad Prism). For all panels: *$P \leq 0.05$, **$P \leq 0.01$, ***$P \leq 0.001$; NS, not significant

mice were administered with each of the antibodies at a dose of 15 mg per kg and challenged with a lethal dose of X31. At 3 and 6 dpi, lungs were collected from the animals and viral titers were measured by plaque assays (Fig. 4f, g). No significant differences were detected between neutrophil-depleted and non-depleted animals for each antibody at both 3 and 6 dpi. Our results demonstrate that neutrophils do not significantly contribute to the in vivo protection mediated by nonNAbs.

**Human broadly reactive nonNAbs protect mice via AMφ.** To confirm that our findings are relevant in the context of human IAV infection, we repeated the aforementioned AMφ depletion experiment using two human nonNAbs 5E01 and 5D06. MAbs 5E01 and 5D06 were recently identified as group 2 HA broadly reactive antibodies with no in vitro neutralizing activity[10]. However, it was shown that they can protect mice from lethal H7N9 infection via Fc–FcγR interactions[10]. To investigate whether AMφ contribute to the protection mediated by 5E01 and 5D06, AMφ were depleted by clodronate liposome and nonNAbs were administered IP in mice as previously described. Mice were then challenged with X31 or H7N9 virus. Non-depleted mice that received 5E01 or 5D06 were protected from the lethal challenge of X31 (Fig. 5a, b) or H7N9 (Fig. 5c, d), whereas mice depleted of AMφ completely lost protection and showed weight loss comparable to control IgG groups. Here, we demonstrate that protection by human broadly reactive nonNAbs require AMφ—indicating that the mechanism by which murine and human nonNAbs protect are similar in mice.

**BNAbs rely on AMφ for protection in vivo at suboptimal doses.** Recent work had demonstrated the importance of Fc–FcγR

interactions for the in vivo protection by HA stalk-specific bNAbs[14]; however, the contributions of which innate immune cell population involved remain undefined. In light of our current findings, we sought to investigate whether AMφ may also play a role in mediating protection by bNAbs. We passively transferred two stalk-specific bNAb 9H10 (pan-H3/H10) or 6F12 (pan-H1)[32] at two doses (15 mg per kg and 5 mg per kg) into AMφ-depleted mice. Following lethal IAV challenge, the AMφ-depleted mice that received 15 mg per kg of 6F12 showed similar morbidity as non-depleted animals (Fig. 6a, c), consistent with previous results where 9H10 was used as a positive control (Fig. 3a). In mice that received a suboptimal dose (5 mg per kg) of 9H10 or 6F12, morbidity was significantly increased in AMφ-depleted groups compared to the non-depleted groups (Fig. 6b, d), indicating that AMφ conferred maximum protection at suboptimal doses of HA stalk-binding bNAbs.

**NonNAb-mediated protection is dependent on FcγR engagement.** The results above clearly demonstrate that AMφ can cooperate with nonNAbs to clear IAV infection in vivo, we therefore turned our attention to investigating whether this protection is Fc–FcγR interaction dependent. We generated mouse IgG2a WT and IgG2a D265A versions of nonNAb FEE8 and 5E01 (chimeric human F(ab′) with murine Fc). The D265A is a mutation on the murine Fc region that abrogates the binding of Fc to FcγRs[33]. This mutation does not affect the binding of FEE8 or 5E01 to HK/68 H3 as shown by ELISA (Fig. 7a, b). As expected, mice that received FEE8 or 5E01 IgG2a WT antibodies were protected against lethal X31 infection, whereas mice treated with D265A antibodies lost protection and showed similar weight

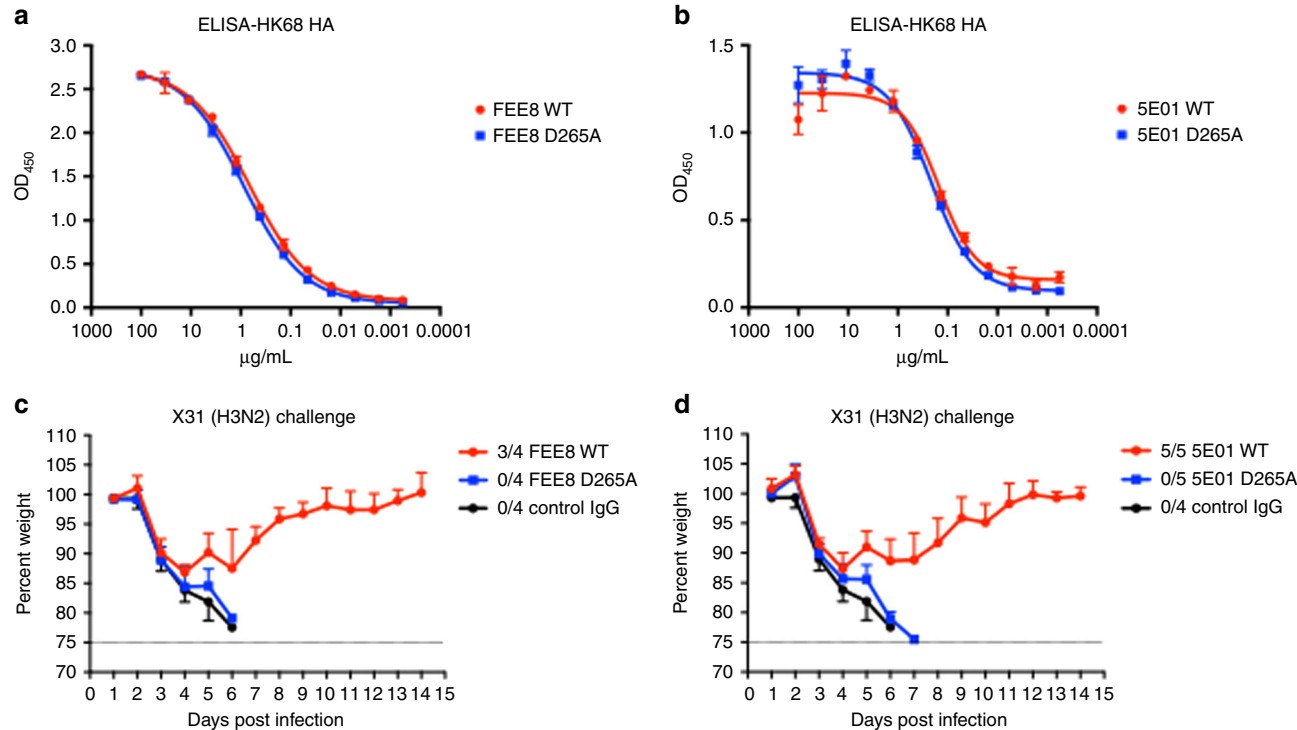

**Fig. 7** In vivo protection by nonNAbs is dependent on Fc-receptor engagement. The variable regions of FEE8 and 5E01 were cloned into murine IgG2a WT or murine IgG2a D265A backbone, respectively. ELISA assays were performed using recombinant HK/68 H3 HA to assess the binding affinities of (**a**) FEE8 WT and FEE8 D265A, or (**b**) 5E01 WT and 5E01 D265A. **c**, **d** BALB/c mice were administered IP with 6 mg per kg of each antibody and then challenged IN with 5 mLD$_{50}$ of X31. A mAb against GST was used as an IgG control. The ratios in the figure legends indicate the number of animals that survived challenge over total number of animals per group ($n = 4$ or 5). Values represent mean ± SD

loss as control IgG group (Fig. 7c, d). Therefore, Fc–FcγR interactions are important for in vivo protection by HA-specific nonNAbs.

**NonNAbs and bNAbs can activate AMφ effector functions.** AMφ play a critical role in initiating the immune responses to inhaled pathogens. Upon virus infection, they can be stimulated to engulf antibody-bound viruses and secrete pro-inflammatory molecules that recruit other effector cells to the site of infection[34]. Murine AMφ express all four FcγRs including three activating receptors and one inhibitory receptor[35]. Therefore, they can be activated by immune complexes to perform effector functions. To investigate whether nonNAbs and bNAbs can stimulate AMφ to secrete pro-inflammatory molecules in vitro, we designed an assay whereby A549 cells were initially infected with IAV. Primary AMφ isolated from mice were then added to interact with antigen-bound antibodies. Supernatants were collected and analyzed for cytokine/chemokine production. Consistent with our previous in vivo results, nonNAb 2B9 and FEE8 were capable of potently stimulating AMφ to release pro-inflammatory cytokines/chemokines compared to control IgG (Fig. 8a). BNAbs 9H10 and 6F12 were also able to elicit cytokine/chemokine release of AMφ. To determine if this observed activation is dependent on Fc–FcγR interactions, we added anti-CD16/CD32 antibodies to block FcγRs on the surface of AMφ. Cytokine/chemokine induction of both nonNAbs and bNAbs was inhibited. Therefore, the activation of AMφ by nonNAbs and bNAbs is dependent on engagement of FcγRs.

AMφ are professional phagocytes highly adept at ingesting and destroying inhaled pathogens. To examine the capability of nonNAbs and bNAbs in mediating phagocytosis, we performed an in vitro bead-based phagocytosis assay. First, fluorescent neutravidin beads were conjugated with HK/68 HA proteins. The mAb 2B9, FEE8, or 9H10 was then incubated with HA-coated beads and the mixture was added to THP-1 cells. After incubation, samples were analyzed by flow cytometry and phagocytic scores were calculated. As seen in Fig. 8b, each antibody was capable of inducing antibody-dependent cell-mediated phagocytosis (ADCP) in a dose-dependent manner, whereas IgG control did not trigger phagocytosis. We next monitored the phagocytic activity of these antibodies ex vivo. Purified preparations of X31 were initially labeled with a pH-sensitive fluorescent dye and then incubated with mAb 2B9, FEE8, 9H10, or control IgG (10 μg per mL) before intranasal infection of naive BALB/c mice. Two hours after infection, lungs were collected and flow cytometry was used to assess low pH-activated fluorescent signal (internalized virus particles) in AMφ. We found that mAb 2B9, FEE8, and 9H10 significantly increased the phagocytosis efficacy of AMφ when compared to IgG control (Fig. 8c). In this case, no significant differences in efficacy were detected between nonNAbs and bNAbs. Taken together, our results demonstrate that both nNAbs and bNAbs are able to activate AMφ to establish a pro-inflammatory environment and phagocytose opsonized pathogens, which may serve as mechanisms for the observed in vivo protection.

## Discussion

A broad-spectrum influenza vaccine should confer protection against divergent influenza virus strains and subtypes[36, 37]. While a strong induction of bNAbs to conserved regions is ideal, the polyclonal response elicited by HA will invariably include nonNAbs[10]. We and others have recently described several

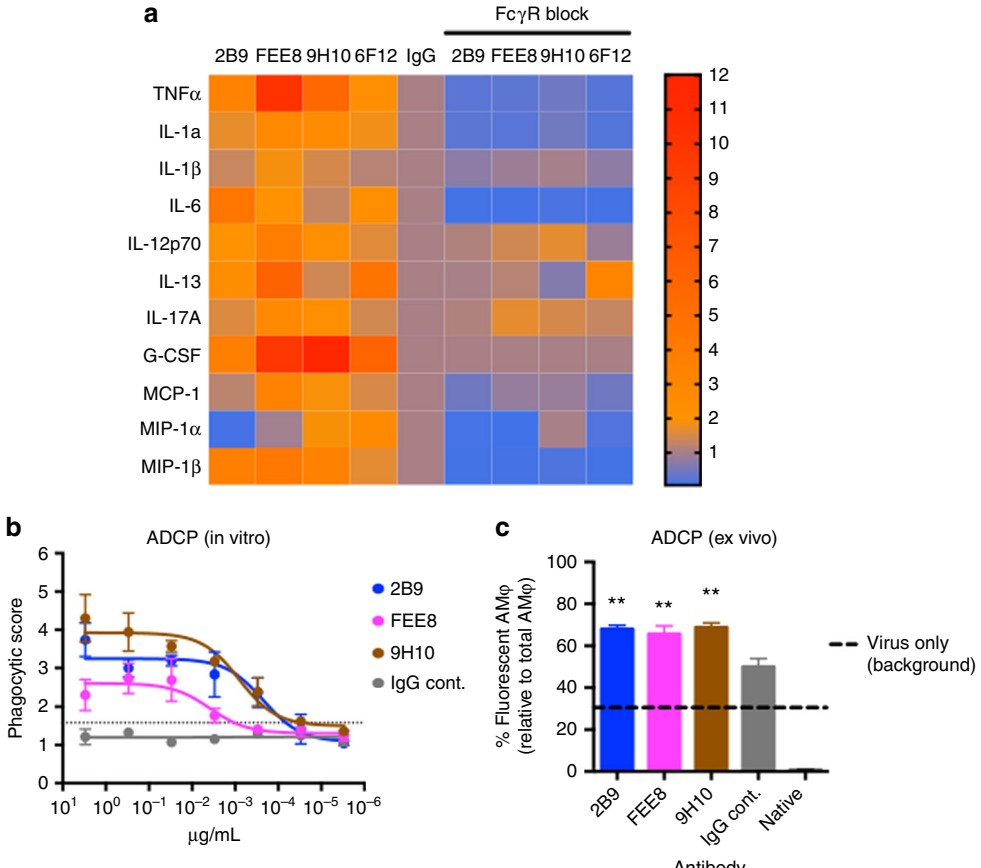

**Fig. 8** NonNAb and bNAb are capable of activating AMφ effector functions. **a** Mouse primary AMφ isolated from BALF were incubated with 50 μg per mL of indicated mAb and X31-infected A549 cells. Seventy-two hours later, supernatants were collected and cytokine/chemokine levels were measured by multiplex bead array assay. The heat map (generated by GraphPad Prism) represents fold changes of cytokine and chemokine levels in supernatants from indicated treatments. Values are normalized to corresponding IgG controls ($n = 3$). **b** An in vitro ADCP assay was performed using THP-1 cells. HA-coated fluorescent beads were incubated with each antibody before added to THP-1 cells. Samples were analyzed by flow cytometry and the phagocytic score was determined. **c** Purified preparations of X31 labeled with a pH-sensitive fluorescent dye were incubated with mAb 2B9, FEE8, or 9H10 before administered intranasally into naive BALB/c mice. Two hours after infection, lungs were collected and internalized fluorescent-labeled viral particles in AMφ were assessed by flow cytometry. The percentage of AMφ with internalized virus particles over total AMφ was calculated. Values represent mean ± SD; the experiment was done in triplicates. One-way ANOVA and Dunnett's multiple comparisons tests were used to determine statistical significance (GraphPad Prism). For all panels: *$P \leq 0.05$, **$P \leq 0.01$, ***$P \leq 0.001$; NS, not significant.

nonNAbs that target the HA and require Fc–FcγR interactions for optimal protection in vivo[9–11]. While a number of studies have highlighted the mechanisms through which influenza-specific antibodies engage FcγRs[38, 39], the contribution of NK cells in conferring protection through antibody-dependent cellular cytotoxicity (ADCC)[40] and its importance as a correlate of protection[41–43] is not known. In fact, the innate immune cell(s) responsible for mediating Fc-dependent effector functions during an acute infection with influenza virus remains to be fully identified. To address this question, we utilized murine mAbs with no in vitro neutralizing activity to determine their mechanisms of action in vivo. We demonstrate that AMφ are critical for conferring optimal homologous and heterologous protection by murine nonNAbs in vivo (Fig. 3).

Our findings with murine nonNAbs led us to ask whether human nonNAbs would exhibit a similar dependence on AMφ for protection against influenza virus. We chose two human mAbs, 5E01 and 5D06[10], to assess their mechanisms in vivo. Similar to murine nonNAbs, 5E01 and 5D06 required AMφ for protection against lethal challenges of H3 or H7 IAV (Fig. 5). Moreover, the protection mediated by both murine and human

nonNAbs rely on Fc–FcγR interactions (Fig. 7). Interestingly, we also show that HA stalk-specific bNAbs require AMφ for optimal protection against influenza. At high concentrations, bNAbs do not require Fc–FcγR interactions for protection, but would necessitate FcγR engagement at lower concentrations[14]. Here, we demonstrate that stalk-specific bNAbs significantly lose protective efficacy in clodronate-treated mice at lower concentrations (5 mg per kg) (Fig. 6), indicating that the effector cells involved in Fc-mediated protection demonstrated by previous studies are most likely AMφ[9, 14].

Resident AMφ produce low levels of cytokines, and suppress the induction of the innate and adaptive immune response. However, upon an acute lung injury, AMφ respond by producing inflammatory cytokines and become phagocytic[44, 45]. Analysis of the cytokine profile from BALF in mice indicates that nonNAb-treated mice had higher amounts of pro-inflammatory cytokines/chemokines in the lungs than IgG control at 3 dpi (Fig. 2). This is consistent with what was detected in the degree of cellular infiltration in the lungs of nonNAb-treated mice at 3 and 6 dpi. While further work may be required to fully define the kinetics of antibody-dependent inflammatory response in these

sets of experiments, we hypothesize that the higher inflammatory cytokine response in the lungs of nonNAb-treated mice at 3 dpi may facilitate the clearance of virus detected at 6 dpi. Of note, we hypothesize that the lack of a strong inflammatory response in 9H10-treated mice is attributed to the neutralizing property of the bNAb in vivo negating the requirement for FcγR engagement. Along with inflammation, FcγR engagement can also trigger phagocytosis, which was demonstrated in ex vivo and in vitro experiments. While the main mechanism by which these antibodies and AMφ interact to clear virus remains to be fully elucidated, our data suggest that induction of an appropriate inflammatory cytokine response and Fc-mediated phagocytosis of immune complexes are involved.

The contribution of antibody affinity in protection is potentially important as we have previously demonstrated that high-affinity nonNAbs can protect against H7 influenza virus challenge with minimal weight loss (11). Nevertheless, here we show that even a high-affinity nonNAb, 1H5 (11), still requires AMφ for optimal protection (Supplementary Fig. 3C). Of note, the disparity in the affinity measurement of FEE8 between bio-layer interferometry (BLI) and ELISA can be confounding. On the basis of our passive transfer studies, we hypothesize that affinity as measured by BLI, which reflects an equilibrium dissociation constant ($K_d$) is a more accurate determinant of protection than the half maximal effective concentration (EC50) value provided by an ELISA assay.

The findings in this study suggest AMφ play a crucial role in Fc-mediated responses during an acute influenza infection in the mouse model—whether or not human AMφ function similarly remains to be determined. The lack of homologous FcγR equivalents expressed on either murine and human innate cells makes it difficult for direct comparisons; however, their functionally related FcγRs and expression patterns may provide clues. We know that there is crosstalk between human IgG isotypes and murine FcγRs. Specifically, the human IgG1 isotype used in our studies has been shown to bind all four murine FcγRs and is able to potently induce Fc effector function[46]. In contrast to neutrophils and NK cells, both murine and human macrophages express a near complete set of activating FcγRs, making them ideal candidates of mediating Fc-dependent processes[47]. This certainly does not preclude the involvement of neutrophils or NK cells in Fc-dependent effector functions in vivo as recent observations suggest that bnAbs have the ability to induce phagocytosis by neutrophils and ADCC by NK cells in vitro[14, 48, 49]. Targeting different innate immune cells, we have determined that migratory inflammatory cells (i.e. neutrophils and NK cells) play a minimal role in mediating Fc effector functions compared to resident cells, such as AMφ. Given that influenza virus can cause lower respiratory infections, it is conceivable that the locale of resident AMφ, with the proper FcγRs, have a preeminent role in participating in antibody-mediated activities. Our work reveals a link between AMφ and broadly binding HA-specific antibodies in protection, and moreover, highlights the roles innate cells have in facilitating an immune response against divergent influenza viruses.

## Methods

**Cells and viruses**. Human embryonic kidney 293 T cells (American Type Culture Collection; ATCC) (cat. no. CRL-1573), Madin-Darby Canine Kidney (MDCK) (ATCC; cat. no. CCL-34) cells, and adenocarcinomic human alveolar basal epithelial cells (A549) (ATCC; cat. no. CCL-185) were grown in Dulbecco modified Eagle medium (DMEM, Gibco) supplemented with 10% fetal bovine serum (FBS) (HyClone) and 100 U/mL of penicillin and streptomycin (Gibco). THP-1 (ATCC) cells (ATCC; cat. no. TIB-202) were maintained in RPMI-1640 (Gibco) supplemented with 10% FBS and 100 U/mL of penicillin and streptomycin (Gibco). A/Shanghai/1/13, X31, and X79 are reassortant viruses that have six internal proteins of A/Puerto Rico/8/34 expressing the HA and NA of the indicated viruses:

X31 (HA and NA of HK/68, H3N2), X79 (HA and NA of A/Philippines/2/82, H3N2), and H7N9 (HA and NA of A/Shanghai/1/13) that have been passaged and grown at the Icahn School of Medicine at Mount Sinai. Other viruses utilized in the paper are: A/Victoria/361/11 (Vic/11) (H3N2), A/Perth/16/09 (H3N2), and A/ Netherlands/602/2009 (NL09) (H1N1) (these viruses were obtained from the Centers for Disease Control and Prevention). All the viruses above were grown in 10-day-old specific-pathogen-free embryonated chicken eggs (Charles River Laboratories, Inc.).

**Antibodies**. MAbs 9H10 (IgG2a)[15], 6F12 (IgG2a)[32], 22A6 (anti-GST mAb) (IgG2b) (29), 5E01, and 5D06[10] have been described previously. Hybridomas 2C10 (IgG2a), 2B9 (IgG2a), and FEE8 (IgG2a) were produced by sequential infection of a female BALB/c mice (6–8 weeks old) (Jackson Laboratories, Inc) with sublethal doses of X31, Vic/11, and X79 in 3-week intervals. Three weeks after the last sublethal infection, the mouse was boosted IP with a purified preparation of A/Perth/16/09 (H3N2) virus. The spleen was collected and dissociated into single-cell suspension in serum-free DMEM 3 days after the boost. Fusion of splenocytes and SP2/0 myeloma cells (ATCC; cat. no. CRL-1581) was performed using poly-ethylene glycol (Sigma-Aldrich, Inc.). Fused cells were grown for 10–12 days. Cell colonies were picked and grown for screening. MAb 22A6 is used as a control IgG in all experiments. The IgG2b isotype was previously shown to have equivalent affinities for FcγRIII and FcγRIV, which are the chief FcγRs involved in Fc-mediated responses in mice[50, 51].

Supernatants were collected from hybridoma cultures and filtered prior to IgG purification. IgG antibodies were purified by a gravity flow column containing protein G-Sepharose 4 Fast Flow (GE Healthcare). The supernatants were passed through the column once and the resins were washed with PBS three times. The antibodies were eluted with 0.1 M glycine (pH 2.7) and the eluate was quickly neutralized with 2 M Tris-HCl buffer (pH 10). Antibodies were concentrated using Amicon Ultra centrifugal filter units (Millipore) with a cutoff of 30 kDa molecular mass.

**ELISA**. ELISA assays were performed on 96-well plates as described previously[52]. Briefly, plates were coated with 2 μg per mL of purified recombinant HA protein overnight in bicarbonate-carbonate coating buffer (100 mM, pH 9.6). On the day of assay, the plates were blocked with 5% non-fat milk for 1 h. Antibodies diluted in 1% milk (the starting concentration of each mAb is at 100 μg per mL and serially diluted threefold) were added to the plates and incubated for 1 h. Plates were washed three times with PBS-0.1% Tween-20 (PBST). Secondary sheep anti-mouse IgG-horseradish peroxidase (HRP; GE Healthcare) was incubated on plates at 1:5000 dilution for 1 h. Plates were washed with PBST three times before adding HRP substrate (Sigmafast OPD; Sigma-Aldrich). Reactions were stopped by addition of 3 M HCl, and the optical densities were read at 490 nm on a Synergy 4 (Bio-Tek) plate reader. A non-linear curve was generated using GraphPad Prism and the half-maximal effective binding concentration (EC50) was calculated.

**Microneutralization assay**. Microneutralization assays were performed as described previously[53]. Briefly, mAbs (starting concentration of each mAb is at 100 μg per mL and serially diluted threefold) and viruses (1000 plaque-forming units) were pre-incubated at 37 °C for 1 h. After incubation, the mixture was added to MDCK cells and incubated at 37 °C for 1 h to allow for adsorption. After washing with PBS three times, the plates were re-incubated at 37 °C with infection medium containing equivalent concentrations of diluted antibody supplemented with 1 μg per mL of tosyl phenylalanyl chloromethyl ketone (TPCK)-treated trypsin (Sigma). Eighteen hours later, the cells were fixed with 80% acetone and then stained for NP using a primary biotinylated antibody (EMD Millipore; cat. no. MAB8257B) (1:2000) and a secondary streptavidin conjugated to HRP antibody (EMD Millipore; cat. no. 18-152) (1:3000) to quantify virus replication[53].

**Western blots**. Recombinant HAs and NAs of HK/68 were produced by a baculovirus system in BTI-TN5B1-4 cells (ThermoFisher Scientific; cat. no. B85502) as described previously[54]. The HA was cleaved using 1 μg per mL trypsin at 37 °C for 20 min, and cleaved HA and NA proteins were then boiled at 100 °C for 5 min in loading buffer containing SDS. Samples were resolved on 4–20% SDS-PAGE gels under non-reducing conditions and transferred to polyvinylidene fluoride membranes. The membranes were then blocked with 5% non-fat milk for 1 h and incubated with either an anti-polyhistidine (His) antibody (HA and NA proteins are His-tagged) (Sigma-Aldrich; cat. no. H1029) diluted at 1:2000, 2C10, 2B9, or FEE8 (diluted to 5 μg per mL) for 18 h to allow for binding. After washing with PBST three times, the blots were incubated with HRP-conjugated secondary antibodies at 1:10,000 dilution and developed by an enhanced chemiluminescence method according to the manufacturer's protocol (Pierce). MAbs XY102 and 12D1 (diluted to 5 μg per mL) (29) were used to detect the HA1 and HA2 of HK/68 HA, respectively.

**Biolayer interferometry**. Biolayer interferometry assays were performed to determine $K_d$ values by using an Octet RED instrument (ForteBio, Inc.). Purified mAbs were loaded onto AMC (Anti-mIgG Fc Capture) biosensors (ForteBio, Inc.) in 1× kinetics buffer (1× PBS (pH 7.4), 0.01% BSA, 0.002% Tween-20) for 3 min.

For the measurement of $k_{on}$, the association was measured for 3 min by exposing the sensors to seven concentrations of purified HA diluted in 1× kinetics buffer. For the measurement of $k_{off}$, the dissociation was measured for 3 min in 1× kinetics buffer. $K_d$ values were calculated as the ratios of $k_{off}$ to $k_{on}$.

**HI assay**. Assays were performed according to standard procedures, as described previously[52]. Briefly, sera were inactivated with TPCK-trypsin at 56 °C for 30 min viruses, and antibodies were mixed and incubated at room temperature for 30 min. Chicken red blood cells were then added to each well. Plates were incubated at 4 °C for 30 min before reading. Plates were scored for the number of wells exhibiting HAI activity.

**Passive transfer experiment in mice**. Groups of female BALB/c mice ($n = 5$) (The Jackson Laboratory) aged 6–8 weeks received mAbs IP. Five hours later, mice were challenged IN with 5 mLD$_{50}$ of X31, X79, or A/Shanghai/1/13 virus. Mice were then monitored daily for weight loss and survival for 14 days. For consistency, mice which lost >20% (as in previous H7N9 studies) (11) or 25% (as in previous H3N2 studies) (15) were sacrificed following the respective virus challenge.

To determine viral titers in the lung, mice ($n = 3$) were sacrificed at 3 or 6 dpi. Lungs were collected and homogenized (BeadBlaster 24; Benchmark), and stored at −80 °C until titered. Viral lung titers were assessed by plaque assays on MDCK cells. Prism was used to graph the results and calculate statistical significance between groups using two-way analysis of variance (ANOVA).

**Innate immune cell depletion studies in mice**. Immune cells including AMφ, neutrophils, and NK cells were depleted in vivo using different methods. For the depletion of AMφ, 50 μl of liposome containing clodronate (ClodronateLiposomes. com) were given IN to BALB/c mice on day −2 and 0. PBS liposome was used as a control. For neutrophil depletion, BALB/c mice were injected IP with 0.25 mg of anti-Ly-6G mAb (1A8; BioXCell) once for every 2 days. PBS was used as a control. For NK cell depletion, C57BL/6J mice received 0.25 mg PK136 mAb (hybridoma purchased from ATCC; cat. no. HB-191) IP on day −2, 0, and 5. PBS was used as a control.

**Bronchoalveolar lavage lung tissue processing and flow cytometry**. BALF was obtained by lavaging the lung with 1 mL of 1× PBS. The BALF was centrifuged at 300×g for 10 min. Cell pellets were collected for flow cytometry analysis. Cytokine and chemokine levels in the cell-free BALF were analyzed using a Bio-Plex Pro Mouse Cytokine 23-plex assay (Bio-Rad Inc.). The assay was performed according to the manufacturer's protocol and was read on a Luminex MAGPIX platform. For each bead set, >50 beads were collected. The median fluorescence intensity of those beads was recorded and the data were analyzed using the Milliplex software. Fold induction was normalized to IgG-treated groups and a heat map was generated using GraphPad Prism.

**Flow cytometry analysis of innate immune cells**. Whole lungs were removed and chopped with razor blades, incubated with type IV collagenase (Worthington) at 37 °C for 40 min, then homogenized through a 70-μm cell strainer (Falcon). Remaining red blood cells were lysed using 1× red blood cell lysis buffer (BD Biosciences). Cells were stained with Fixable Viability Dye eFluor® 455 (eBioscience). Anti-mouse immunophenotyping antibodies were diluted in FACS buffer (3% FBS, 2 mM EDTA, 1× PBS) to 5 μg per mL along with Fc block (anti-mouse CD16/CD32; 5 μg per mL, BD), and cells were stained for 30 min on ice in three groups (AMφ: CD45 (30-F11; BD), CD11c (HL3; BD), CD11b (M1/70; BD), and Siglec-F (E50-2440; BD); monocyte and neutrophil: Ly-6C (AL-21; BD), Ly-6G (1A8; BD), and CD11b (M1/70; BD); dendritic cell: CD45 (30-F11; BD), CD11c (HL3; BD), CD11b (M1/70; BD), MHC II (M5/114.15.2; BD), and CD103 (M290; BD); NK cells: CD3 (17A2; BD), NK-1.1 (PK136; BD), and CD49b (DX5; BD)). After the staining, cells were washed twice with FACS buffer and then fixed in 2% para-formaldehyde in FACS buffer for 15 min. Cell numbers were counted using AccuCount Fluorescent Particles (Spherotech). All data were collected on an LSR II flow cytometer (BD) and analyzed using FlowJo software.

**Alveolar macrophage reconstitution studies**. Lungs were collected and processed as described above from healthy 6- to 8-week-old female C57BL/6J mice (The Jackson Laboratory) and alveolar macrophages were bulk-sorted using a BD FACSAria II cell sorter (BD Biosciences, Inc.) into serum-free 1× DMEM medium and washed once with 1× PBS. Alveolar macrophages were reconstituted to 200,000 cells per 50 μL (1× PBS). Six- to 12-week-old female GM-CSF knockout mice, Csf2$^{tm1Mlg}$ (The Jackson Laboratory), were then reconstituted with 200,000 (in 50 μL) wild-type alveolar macrophages IN on the same day or with 50 μL of PBS. Twenty-four hours post adoptive transfer, GM-CSF knockout mice ($n = 3$ to 4 mice per group) were administered with 15 mg per kg of mAbs IP and challenged IN with 60 p.f.u. per 20 μL of X31. Mice were observed daily for weight change for 14 days.

To confirm successful reconstitution of alveolar macrophages, recipient 6- to 8-week-old female GM-CSF knockout (CD45.2) mice were adoptively transferred (IN) with bulk-sorted alveolar macrophages isolated from donor female B6.SJL (CD45.1) mice, as described above. Three days post adoptive transfer, recipient 6- to 8-week-old female GM-CSF knockout (CD45.2) (104; BD) mice were killed, lung collected, and donor alveolar macrophages (CD45.1) (A20; BD) were identified by flow cytometry analysis (Supplementary Fig. 5) (antibodies were diluted in FACS buffer to 5 μg per mL).

**Ex vivo stimulation of alveolar macrophages**. A549 cells were infected with either X31 (H3N2) or NL/09 (H1N1) at a multiplicity of infection (MOI) of 3 for 18 h in the absence of TPCK-treated trypsin. MAb 2B9, FEE8, 9H10, 6F12, or control IgG (anti-GST mAb) were added at a concentration of 50 μg per mL in 50 μL. Alveolar macrophages from freshly collected BALF of naive mice were subsequently added at a concentration of 1000 cells in 50 μL. The ratio of virus-infected cells to primary alveolar macrophages was 4:1. The FcγR block was a mixture of anti-CD32 and anti-CD16 antibodies (BD Biosciences; cat. no. 553141) (1 μg per million cells in 100 μL). Seventy-two hours later, cell culture supernatants were collected and assessed for cytokine production using a Bio-Plex Pro Mouse Cytokine 23-plex assay as described above. Fold induction was normalized to IgG control and a heat map was generated using GraphPad Prism.

**Bead-based phagocytosis assay**. Assays were performed according to the method described previously[55]. Briefly, biotinylated recombinant HK/68 HA antigen was conjugated to fluorescent neutravidin beads (580/605) overnight at 4 °C. Unconjugated protein was removed by three washes with buffer (1× PBS, 1% BSA). After the last wash, beads were diluted 1:100 in buffer (1× PBS, 1% BSA) and stored at 4 °C. Serial diluted antibodies (starting at 12 μg per mL, 50 μL per well) were incubated with HA-conjugated beads (50 μL per well) in 96-well U-shape plates for 2 h at 37 °C. Later, THP-1 cells were added (20,000 cells per well in total of 100 μL) and incubated for another 12 h at 37 °C before fixation with 3.7% paraformaldehyde. Plates were measured at LSRII flow cytometer (BD Biosciences) using the HTA adaptor. Acquired numbers were analyzed using FCS Express Flow Cytometry Software (DeNovo). Phagocytic score was calculated as follows: (MFI × %ofparentpopulation) ÷ 1,000000. All the samples were analyzed in triplicates. Cutoff is defined as the mean of the negative controls + 3 SD.

**Ex vivo phagocytosis assay**. A purified preparation of X31 was labeled with a pH-sensitive fluorescent dye (pHrodo Green STP ester; ThermoFisher). Briefly, a 10 mM stock solution of the dye was made by adding 75 μL of DMSO to 500 μg pHrodo Green STP ester. Fifty micrograms of a purified preparation of X31 (1 mg per mL) was then added to the reconstituted dye and incubated at room temperature in the dark for 60 min. The labeled virus was then dialyzed against 1× PBS overnight at 4 °C in the dark. Eight micrograms of purified virus preparation were pre-incubated with 100 μg per mL of mAb for 30 min at RT and then were subsequently used to infect 6- to 8-week-old female BALB/c mice IN. Two hours after infection, mice were sacrificed, lungs were collected, and alveolar macrophages were analyzed by flow cytometry as described above. Internalized viral particles in alveolar macrophages were indicated with green fluorescence. Two controls were utilized for this experiment: (i) pHrodo-labeled virus and (ii) pHrodo-labeled virus with control IgG. The percentage of AMφ with internalized virus particles over total AMφ was calculated. The experiment was done in triplicates. One-way ANOVA and Dunnett's multiple comparisons test were used to determine statistical significance (GraphPad Prism).

**Data availability**. The authors declare that the data supporting the findings of this study are available within the article and its Supplementary Information files, or are available from the authors upon request.

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

## Acknowledgements

We thank M. Schotsaert for useful discussions, A. Hirsh for protein production, and V. Gillespie for her expertise in scoring the lung sections. We also acknowledge the use of the Mount Sinai Biorepository and Pathology Center of Research Excellence (CORE) and the Flow Cytometry Shared Resource Facility. This work was supported by Center for Research on Influenza Pathogenesis Grant HHSN272201400008C (to P.P. and F.K.) and HHSN272201400005C (to P.C.W.); National Institute of Allergy and Infectious Diseases Grants P01-AI097092 (to P.P., F.K. and P.C.W.), U19-AI109946 (to P.P., F.K. and P.C.W.), and P01-AI097092-04S1 (to P.E.L.); and the Canadian Institutes of Health Research (M.S.M.). The animal studies performed in this study are in accordance with the Institutional Animal Care and Use Committee of the Icahn School of Medicine at Mount Sinai.

## Author contributions

W.H. and G.S.T. conceived the study. W.H., C.-J.C., C.E.M., J.R.H., C.K.W., P.E.L., M.B.U., V.C., K.W.H. and G.S.T. designed and performed the experiments. C.H. and P.C.W. isolated and characterized the human mAbs. J.K.L., M.S.M., F.K. and P.P. alongside the other authors analyzed the data. W.H. and G.S.T. wrote the manuscript, and all authors reviewed and approved the final version.
