## [Peer Review File · Nature Communications]

Reviewers' comments:

Reviewer #1 (Remarks to the Author):

This is a very straightforward paper that makes one main conceptual point that the Fc region dependent protective effect of non-neutralizing monoclonal antibodies for influenza require alveolar macrophages for optimal protective effect in a mouse model of infection.

The data do seem consistent with the conceptual idea that alveolar macrophages do play a critical role in the effect of Abs in this model.

Exactly how (the mechanism by which) these macrophages contribute to protection in the presence of antibodies is not really fully fleshed out here.

The paper is a little contrarian given most people in the field are studying NK cell assays for correlation with in vivo studies (and often find that these types of antibodies possess activity with NK cell lines or primary cells in vitro), but granted the data showing these effects specifically occur in vivo with restriction to NK cells are limited in the literature.

The manuscript describes the study of three murine mAbs (broad H3) or two human nonNAbs 5E01 and 5D06, or suboptimal dosing with some stem antibodies. Measurement of some murine cytokines/chemokines in mice with protective nonNAbs suggest the mAbs are associated with high "alveolar inflammatory responses" during infection

Depletion in vivo (using IN clodronate-liposome) abrogates protection and increases viral titer. Mab depletion of NK cells or neutrophils have little effect. The effect was Fc dependent (murine Fc D265A variants don't mediate protection)

The weakest part of the paper is the exploration of what macrophages are contributing to this effect, i.e. how are they mediating an inhibitory effect?

Basically, inflammatory factor secretion is noted in alveoli, and exposure of murine macrophages to human origin influenza infected cells causes secretion of soluble factors. Granted, a "pro-inflammatory environment" may be observed, but is this the basis for protection? There are a whole lot of relatively easy experiments to get at macrophage functions that would be informative here.

Specific comments:

Line 111 "were 2-3 logs lower" – should indicate log10, log2 etc, or use another terminology.

Reviewer #2 (Remarks to the Author):

The authors describe three mouse monoclonal antibodies that bind to recombinant H3 HA derived from a set of human H3N2 viruses that were isolated in 1968, 1975, 1982 and 2011. Compared to the HA stem specific mAb 9H10, these monoclonal antibodies bind to recombinant HA with an approximately 100 to 1000 fold lower affinity and they do not exhibit detectable virus neutralizing activity. Prophylactic treatment of BALB/c mice at a dose of 15 mg/kg protected against challenge with X31 or X79 virus. Alveolar macrophages seem to be important for protection by these mAbs as well as for protection by a "suboptimal" dose of stem-specific monoclonal antibodies 9H10 and 6F12. Non-neutralizing HA-specific human monoclonal antibodies also fail to protect mice that have been treated with clodronate-loaded liposomes. Recombinant monoclonal antibodies with D275A in the IgG2a Fc do not protect against X31 challenge. Remarkably, protection by the non-neutralizing antibodies 2B8 and FEE8 correlates with higher levels of pro-inflammatory cytokines on day 3 after challenge and with higher alveolar inflammation on day 6 after challenge.

The involvement of Fc Receptors and alveolar macrophages for protection by (non-neutralizing) antibodies against influenza virus challenge has been reported before (Laidlaw et al., 2013, Huber et al., 2001, El Bakkouri et al., 2011, Song et al., 2011) and phagocytosis by 5E01 and 5D06 (Dunand et al., 2016). The protection by the non-neutralizing monoclonal antibodies described in this study requires a high dose, which, when extrapolated to an adult would correspond to 1 gram of antibody. Even at this dose, the mice lose considerable weight after challenge and virus loads in the lungs remain very high. Finally, it is difficult to imagine how a vaccine that elicits broadly protective, inflammation promoting antibodies can go together with clinical relieve.

Major remarks.

1. The conclusion that AMs are required for protection by the nonNABs against influenza virus challenge relies heavily on the clodronate depletion protocol. However, such a treatment also potentially eliminates part of the interstitial macrophages and DCs in the lungs. In addition, the PBS-loaded liposome control is not neutral and induces some level of inflammation. Figure 4C illustrates this: compared to naïve mice, monocytes and neutrophils are upregulated in the liposome treated animals. A reconstitution experiment is essential to provide convincing evidence that AMs are required for protection by nonNABs. Such a reconstitution after clodronate treatment should be performed with AMs derived from wild type and Fc R knock out mice.
2. The affinity of FEE8 as determined by biolayer interferometry is not in line with the EC50 determined by ELISA for the Philippines 82 recombinant HA: 9H10 and FEE8 have a comparable EC50 for this protein but differ more than 100 fold in the interferometry assay. Which of the two measurements is correct? It is more relevant and important to compare the affinity of the mAbs for influenza A virions and infected cells.
3. The inflammatory cytokine levels should be compared to mock-infected controls. Furthermore, it is unclear how the heat map in figure 2C was generated. Based on the box plots in Figure S2 the levels of IL17A were on average three fold higher in challenged FEE8 treated mice compared to IgG control treated mice. Yet the heat map suggests a 6-8 fold difference. What do the boxplots represent in Fig S2?
4. Figure 8 shows that in vitro, after 72h co-incubation of AMs with infected A549 cells, inflammatory cytokines are produced in an Fc Receptor dependent way. However this is also

the case for the nonNAbs and for 9H10, which is not in line with the in vivo outcome in BAL cytokine levels. There thus may be a disconnect between the pro-inflammatory in vivo results and the role of AMs in protection. In fact, comparison of Figure 1C and Fig 2C allows one to conclude that protection in terms of weight loss, correlates with control of inflammation (although not for the control treated animals). So the question is which cytokine or chemokine is responsible for the disease and could morbidity be reduced by neutralizing e.g. IL1B or IL17A in the nonNAb-treated mice? In addition, is depletion of AMs associated with reduced inflammation in vivo for challenged mice that have been treated with the nonNAbs and a suboptimal dose of 9H10?

5. Figure S3 (X79 challenge) lacks the treatment with 9H10 Ab. In addition, unlike X31, X79 and H7N9 challenge viruses result in accelerated morbidity and death in control IgG treated mice that lack AMs. Therefore, the conclusion in line 155-157 is not correct and the contribution of AMs in protection by nonNAbs appears to be challenge virus strain-dependent.

6. There are discrepancies between some of the experiments. NK cell depletion was performed in Bl/6 mice while most experiments were performed in BALB/c. The ethical endpoints for the H7N9 virus challenge is 20% body weight loss and 25% for the other experiments. So a mouse may be considered alive or death dependent on the predefined endpoint.

Other remarks:

1. Without statistical analysis, the 9H10 control and the results of mice with AMs, figures 3G and 3H and the accompanying text are confusing. What are the other factors?

2. Except for the color codes, panel 3A and 6A are identical.

3. The infected A549 – AM co-culture is poorly described. What was the ratio of infected cells over AMs? Which Fc block antibody was used? NL/09 (H1N1) is mentioned. Where are the results obtained with that virus?

4. The IgG2b control antibody is not ideal: this isotype has a much lower affinity for the mouse FcγR1 receptor than IgG2a.

5. First line of the abstract: the aim of universal influenza vaccine candidates is better defined as “to provide protection against all influenza A and B viruses”. Line 48: the authors show that broadly neutralizing antibodies can protect mice that have been treated i.n. with clodronate or PBS loaded liposomes equally well.

Reviewer #3 (Remarks to the Author):

The MS by He et al studies the role of alveolar macrophages in assisting the partial protection of mice from IAV by HA head binding Non-NAb. Overall the work is well done and fits well with the growing interest in Fc-mediated function of IAV Abs. I have some reservations about the interpretation of the depletion experiments and the proposed FcR interactions.

Comments

The three antibodies, 2B9, 2C10 and FEE8 had relatively low affinity for HA in comparison to

the BnAb control - could this alter the cell types required?

Alveolar inflammation is interpreted as effective immunity but does the alveolar inflammation measured only come from immune responses or does infection contribute – why at day 6 is the Bnab group similar to the IgG control group (about to be euthanased with influenza). The marked histologic changes seem out of kilter with the survival of the animals, although the nonNab groups only barely survive.

The depletion of alveolar macs is done chemically and locally whereas the depletion of NK cells and Neutrophils is done systemically via antibodies. Although I recognize the technical limitations or alternate investigations, I'm not sure these 2 methods make comparing the effect of the different cell types valid.

What is the Fc functionality of the human non-Nabs studied on murine cells?

It would be interesting to know whether CD32 or CD16 are primarily mediating the effects observed. The utility of HA-specific phagocytosis has been recently described (Ana-Sosa-Batiz, Plos One 2016)

Minor comment:

Fig 1b – it would be useful to show the stalk binding BnAb control here.

We thank the editor and reviewers for their insightful suggestions and comments on our manuscript. All of the reviewers' comments have been carefully considered and additional results have been provided to address them. Specifically, in the revised manuscript, we have included 1) adoptive transfer results (Figure 3G and H; Figure S5), which indicate that alveolar macrophages are required for the protection by nonNAbs and 2) *in vitro* and *ex vivo* phagocytosis results (Figure 8B and C), which show that phagocytosis mediated by alveolar macrophages can also contribute to the protection by both non-neutralizing and broadly-neutralizing antibodies. We believe that we have thoroughly addressed each of the concerns raised by the reviewers below and we hope that the revised version of our manuscript will be acceptable for publication in *Nature Communications*.

Reviewers' comments:

Reviewer #1 (Remarks to the Author):

This is a very straightforward paper that makes one main conceptual point that the Fc region dependent protective effect of non-neutralizing monoclonal antibodies for influenza require alveolar macrophages for optimal protective effect in a mouse model of infection.

The data do seem consistent with the conceptual idea that alveolar macrophages do play a critical role in the effect of Abs in this model.

Exactly how (the mechanism by which) these macrophages contribute to protection in the presence of antibodies is not really fully fleshed out here.

The paper is a little contrarian given most people in the field are studying NK cell assays for correlation with *in vivo* studies (and often find that these types of antibodies possess activity with NK cell lines or primary cells *in vitro*), but granted the data showing these effects specifically occur *in vivo* with restriction to NK cells are limited in the literature.

The manuscript describes the study of three murine mAbs (broad H3) or two human nonNAbs 5E01 and 5D06, or suboptimal dosing with some stem antibodies. Measurement of some murine cytokines/chemokines in mice with protective nonNAbs suggest the mAbs are associated with high "alveolar inflammatory responses" during infection.

Depletion *in vivo* (using IN clodronate-liposome) abrogates protection and increases viral titer. Mab depletion of NK cells or neutrophils have little effect. The effect was Fc dependent (murine Fc D265A variants don't mediate protection)

The weakest part of the paper is the exploration of what macrophages are contributing to this effect, i.e. how are they mediating an inhibitory effect?

Basically, inflammatory factor secretion is noted in alveoli, and exposure of murine macrophages to human origin influenza infected cells causes secretion of soluble factors. Granted, a "pro-inflammatory environment" may be observed, but is this the basis for protection? There are a whole lot of relatively easy experiments to get at macrophage functions that would be

informative here.

We hypothesize that Fc engagement by nonNAbs triggers a host of antiviral pathways and processes. Here, we demonstrate that a combination of Fc-mediated inflammation and phagocytosis contributes to the clearance of virus and/or virus infected cells and prevents death. In one set of additional experiments, we performed *in vitro* and *ex vivo* phagocytosis assays and showed that our protective nonNAbs and bNAbs can potently stimulate the phagocytic activity of macrophages. The results are outlined in Figure 8B and C and described in lines 311 to 326.

Specific comments:

Line 111 “were 2-3 logs lower” – should indicate log₁₀, log₂ etc, or use another terminology.

We have now changed it to “were 100-1000 fold lower”. Please see line 121.

Reviewer #2 (Remarks to the Author):

The authors describe three mouse monoclonal antibodies that bind to recombinant H3 HA derived from a set of human H3N2 viruses that were isolated in 1968, 1975, 1982 and 2011. Compared to the HA stem specific mAb 9H10, these monoclonal antibodies bind to recombinant HA with an approximately 100 to 1000 fold lower affinity and they do not exhibit detectable virus neutralizing activity. Prophylactic treatment of BALB/c mice at a dose of 15 mg/kg protected against challenge with X31 or X79 virus. Alveolar macrophages seem to be important for protection by these mAbs as well as for protection by a “suboptimal” dose of stem-specific monoclonal antibodies 9H10 and 6F12. Non-neutralizing HA-specific human monoclonal antibodies also fail to protect mice that have been treated with clodronate-loaded liposomes. Recombinant monoclonal antibodies with D275A in the IgG2a Fc do not protect against X31 challenge. Remarkably, protection by the non-neutralizing antibodies 2B8 and FEE8 correlates with higher levels of pro-inflammatory cytokines on day 3 after challenge and with higher alveolar inflammation on day 6 after challenge.

The involvement of Fc Receptors and alveolar macrophages for protection by (non-neutralizing) antibodies against influenza virus challenge has been reported before (Laidlaw et al., 2013, Huber et al., 2001, El Bakkouri et al., 2011, Song et al., 2011) and phagocytosis by 5E01 and 5D06 (Dunand et al., 2016). The protection by the non-neutralizing monoclonal antibodies described in this study requires a high dose, which, when extrapolated to an adult would correspond to 1 gram of antibody. Even at this dose, the mice lose considerable weight after challenge and virus loads in the lungs remain very high. Finally, it is difficult to imagine how a vaccine that elicits broadly protective, inflammation promoting antibodies can go together with clinical relieve.

Major remarks.

1. The conclusion that AMs are required for protection by the nonNAbs against influenza virus challenge relies heavily on the clodronate depletion protocol. However, such a treatment also potentially eliminates part of the interstitial macrophages and DCs in the lungs. In addition, the PBS-loaded liposome control is not neutral and induces some level of inflammation. Figure 4C

illustrates this: compared to naïve mice, monocytes and neutrophils are upregulated in the liposome treated animals. A reconstitution experiment is essential to provide convincing evidence that AMs are required for protection by nonNAbs. Such a reconstitution after clodronate treatment should be performed with AMs derived from wild type and Fc R knock out mice.

As requested by reviewer #2, we added AM ϕ reconstitution studies. We adoptively transferred 200,000 wildtype AM ϕ from naïve C57BL/6J into GM-CSF knockout mice to see if it can rescue the protection mediated by nonNAbs. The results were added to Figure 3 (Figure 3G and 3H; lines 208 to 219 and 512 to 526). Our adoptive transfer experiment indicated that AM ϕ were required for *in vivo* protection mediated by nonNAbs.

2. The affinity of FEE8 as determined by biolayer interferometry is not in line with the EC50 determined by ELISA for the Philippines 82 recombinant HA: 9H10 and FEE8 have a comparable EC50 for this protein but differ more than 100 fold in the interferometry assay. Which of the two measurements is correct? It is more relevant and important to compare the affinity of the mAbs for influenza A virions and infected cells.

We have addressed the reviewers questions. Please see lines 371 to 378.

3. The inflammatory cytokine levels should be compared to mock-infected controls. Furthermore, it is unclear how the heat map in figure 2C was generated. Based on the box plots in Figure S2 the levels of IL17A were on average three fold higher in challenged FEE8 treated mice compared to IgG control treated mice. Yet the heat map suggests a 6-8 fold difference. What do the boxplots represent in Fig S2?

We thank the reviewer for this important observation. The heat map in Figure 2C was generated using the complete set of raw data. To be consistent with heat map results, we have now remade our box plots using the complete set of raw data. As shown by the new Supplementary Figure S2A, the overall trend that 2B9 or FEE8-treated mice have higher inflammatory responses than 9H10 and IgG control group remains the same.

4. Figure 8 shows that *in vitro*, after 72h co-incubation of AMs with infected A549 cells, inflammatory cytokines are produced in an Fc Receptor dependent way. However, this is also the case for the nonNAbs and for 9H10, which is not in line with the *in vivo* outcome in BAL cytokine levels.

The reviewer's question has been addressed. Please see lines 363 to 366.

There thus may be a disconnect between the pro-inflammatory *in vivo* results and the role of AMs in protection. In fact, comparison of Figure 1C and Fig 2C allows one to conclude that protection in terms of weight loss, correlates with control of inflammation (although not for the control treated animals). So the question is which cytokine or chemokine is responsible for the disease and could morbidity be reduced by neutralizing e.g. IL1B or IL17A in the nonNAb-treated mice? In addition, is depletion of AMs associated with reduced inflammation *in vivo* for challenged mice that have been treated with the nonNAbs and a suboptimal dose of 9H10?

It is interesting to speculate the role of IL-1b or IL-17a, however, we have not performed any experiments directly addressing the role of each cytokine. These are questions we are interested in pursuing in our future research aims. We do know from our other data that the depletion of alveolar macrophages was associated with reduced inflammation in animals that received nonNAbs (lines 170 to 171). Also, we have added to our supplemental figures Figure S2B suggesting that alveolar macrophages are required to induce pro-inflammatory response in nonNAb-treated mice.

5. Figure S3 (X79 challenge) lacks the treatment with 9H10 Ab.

The reviewer's comment has been addressed. Please see lines 174 to 175.

In addition, unlike X31, X79 and H7N9 challenge viruses result in accelerated morbidity and death in control IgG treated mice that lack AMs. Therefore, the conclusion in line 155-157 is not correct and the contribution of AMs in protection by nonNAbs appears to be challenge virus strain-dependent.

The reviewer's comment has been addressed. Please see lines 177 to 179.

6. There are discrepancies between some of the experiments. NK cell depletion was performed in Bl/6 mice while most experiments were performed in BALB/c.

The reviewer's comment has been addressed. Please see lines 236 to 238.

The ethical endpoints for the H7N9 virus challenge is 20% body weight loss and 25% for the other experiments. So a mouse may be considered alive or death dependent on the predefined endpoint.

We have addressed the reviewer's comment. Please see lines 477 to 478.

Other remarks:

1. Without statistical analysis, the 9H10 control and the results of mice with AMs, figures 3G and 3H and the accompanying text are confusing. What are the other factors?

We've moved Figure 3G and 3H to Figure 4A and 4B, respectively. Additionally, we rewrote the paragraph to better reflect the data for Figure 4A and 4B (lines 223 to 231). We performed statistical analysis on Figure 4B.

2. Except for the color codes, panel 3A and 6A are identical.

We are aware of this and we have explained it in the figure legend of Figure 6A. We duplicated the panel for the purpose of ease of comparison.

3. The infected A549 – AM co-culture is poorly described. What was the ratio of infected cells over AMs? Which Fc block antibody was used? NL/09 (H1N1) is mentioned. Where are the results obtained with that virus?

We thank the reviewer for this suggestion and have now included the information in the Materials and Methods session (“Ex vivo stimulation of alveolar macrophages”). Please see lines 527 to 535. MAb 6F12 was described in original manuscript in Figure 8A (lines 275 to 280).

4. The IgG2b control antibody is not ideal: this isotype has a much lower affinity for the mouse FcγR1 receptor than IgG2a.

The reviewer’s comment has been addressed. Please see lines 421 to 423.

5. First line of the abstract: the aim of universal influenza vaccine candidates is better defined as “to provide protection against all influenza A and B viruses”.

We have changed the first line of the abstract accordingly. Please see lines 44 to 45.

Line 48: the authors show that broadly neutralizing antibodies can protect mice that have been treated i.n. with clodronate or PBS loaded liposomes equally well.

The reviewer’s comment has been addressed. Please see line(s) 160, 174 to 175, 273 to 208 and 348 to 353.

Reviewer #3 (Remarks to the Author):

The MS by He et al studies the role of alveolar macrophages in assisting the partial protection of mice from IAV by HA head binding Non-NAb. Overall the work is well done and fits well with the growing interest in Fc-mediated function of IAV Abs. I have some reservations about the interpretation of the depletion experiments and the proposed FcR interactions.

Comments

The three antibodies, 2B9, 2C10 and FEE8 had relatively low affinity for HA in comparison to the BnAb control - could this alter the cell types required?

We have added new data to address the reviewer’s question. Please see Figure S3C in the supplementary figure and lines 181 to 187 and 371 to 378.

Alveolar inflammation is interpreted as effective immunity but does the alveolar inflammation measured only come from immune responses or does infection contribute – why at day 6 is the Bnab group similar to the IgG control group (about to be euthanased with influenza). The

marked histologic changes seem out of kilter with the survival of the animals, although the nonNab groups only barely survive.

We have addressed the reviewer's comments. Please see lines 135 to 140.

The depletion of alveolar macrophages is done chemically and locally whereas the depletion of NK cells and Neutrophils is done systemically via antibodies. Although I recognize the technical limitations or alternate investigations, I'm not sure these 2 methods make comparing the effect of the different cell types valid.

We appreciate the reviewer's concerns and addressed it in lines 392 to 394.

What is the Fc functionality of the human non-Nabs studied on murine cells?

We have addressed this in lines 384 to 386.

It would be interesting to know whether CD32 or CD16 are primarily mediating the effects observed. The utility of HA-specific phagocytosis has been recently described (Ana-Sosa-Batiz, Plos One 2016)

We performed two types of assays that indicate that phagocytosis plays a role in the Fc-mediated responses we observed and the additional data are shown in Figures 8B and 8C. Also, please see lines 311 to 326 and 536 to 561.

Minor comment:

Fig 1b – it would be useful to show the stalk binding BnAb control here.

We have already included one stalk-binding bNAb (12D1) here as a positive control, as shown in Figure 1B, left panel.

REVIEWERS' COMMENTS:

Reviewer #2 (Remarks to the Author):

The authors have responded well to the remarks of the reviewers. The additional experiments have strengthened the conclusions of the paper.

Reviewer #3 (Remarks to the Author):

I am satisfied with the response.